# Unveiling Early Signs of Preclinical Alzheimer’s Disease Through ERP Analysis with Weighted Visibility Graphs and Ensemble Learning

**DOI:** 10.3390/bioengineering12080814

**Published:** 2025-07-29

**Authors:** Yongshuai Liu, Jiangyi Xia, Ziwen Kan, Jesse Zhang, Sheela Toprani, James B. Brewer, Marta Kutas, Xin Liu, John Olichney

**Affiliations:** 1Department of Computer Science, University of California, Davis, CA 95616, USA; zkan@ucdavis.edu (Z.K.); xinliu@ucdavis.edu (X.L.); 2Center for Mind and Brain and Neurology Department, University of California, Davis, CA 95618, USA; jixia@ucdavis.edu (J.X.); jmolichney@ucdavis.edu (J.O.); 3Department of Computer Science, University of Southern California, Los Angeles, CA 90089, USA; jessez@usc.edu; 4Department of Neurology, Division of Epilepsy, University of California, Davis, CA 95817, USA; sctoprani@ucdavis.edu; 5Departments of Radiology and Neurosciences, University of California, San Diego, CA 92037, USA; jbrewer@health.ucsd.edu; 6Department of Cognitive Science, University of California, San Diego, CA 92037, USA; mkutas@ucsd.edu

**Keywords:** Alzheimer’s disease, EEG, event-related potential, weighted visibility graph, ensemble learning

## Abstract

The early detection of Alzheimer’s disease (AD) is important for effective therapeutic interventions and optimized enrollment for clinical trials. Recent studies have shown high accuracy in identifying mild AD by applying visibility graph and machine learning methods to electroencephalographic (EEG) data. We present a novel analytical framework combining Weighted Visibility Graphs (WVG) and ensemble learning to detect individuals in the “preclinical” stage of AD (preAD) using a word repetition EEG paradigm, where WVG is an advanced variant of natural Visibility Graph (VG), incorporating weighted edges based on the visibility degree between corresponding data points. The EEG signals were recorded from 40 cognitively unimpaired elderly participants (20 preclinical AD and 20 normal old) during a word repetition task. Event-related potential (ERP) and oscillatory signals were extracted from each EEG channel and transformed into a WVG network, from which relevant topological features were extracted. The features were selected using *t*-tests to reduce noise. Subsequent statistical analysis reveals significant disparities in the structure of WVG networks between preAD and normal subjects. Furthermore, Principal Component Analysis (PCA) was applied to condense the input data into its principal features. Leveraging these PCA components as input features, several machine learning algorithms are used to classify preAD vs. normal subjects. To enhance classification accuracy and robustness, an ensemble method is employed alongside the classifiers. Our framework achieved an accuracy of up to 92% discriminating preAD from normal old using both linear and non-linear classifiers, signifying the efficacy of combining WVG and ensemble learning in identifying very early AD from EEG signals. The framework can also improve clinical efficiency by reducing the amount of data required for effective classification and thus saving valuable clinical time.

## 1. Introduction

Alzheimer’s disease (AD) presents a growing challenge to global public health due to its impact on cognitive function and quality of life. As the population ages, the prevalence of AD is expected to increase, underscoring the need for effective early intervention. Detecting AD at its earliest stages, particularly in the preclinical phase (preAD), is critical for timely interventions and dementia prevention.

AD is characterized by a sequence of biological events that begins years before clinical symptoms [1]. Amyloid-β (Aβ) deposition on PET scans or low Aβ levels in Cerebrospinal fluid (CSF) are considered early indicators of AD in normal older individuals, who may be classified as having preclinical AD (preAD) [2]. Aβ peptides are known to influence synaptic activity with inhibitory effects at post-synaptic sites and excitatory effects at pre-synaptic sites [3,4]. Their pathological accumulation disrupts synaptic transmission [5], alters network-level neuronal activity [3,4], and causes synaptic loss [6,7]. Synaptic abnormalities may occur before amyloid plaque deposition [3,8] and are central to the pathophysiology of AD in preclinical and symptomatic stages [9]. Synaptic loss is strongly associated with the severity of clinical symptoms [10], underscoring the value of identifying biomarkers that can detect early synaptic dysfunction.

EEGs capture summated excitatory and inhibitory postsynaptic potentials [11] and provide a non-invasive measure of synaptic and network functions. EEG-derived measures, such as event-related brain potentials (ERPs), are sensitive to subtle brain changes in early AD [12,13], even in the preclinical stage [14,15]. With its high temporal resolution, EEG is well-suited to track changes in cognitive processes such as memory. Using a word repetition paradigm that elicits language- and memory-related brain activity, our group has identified several ERP measures that distinguish individuals across different AD stages from healthy controls [15,16,17,18,19,20]. For example, the N400 (linked to semantic processing) and the P600 or ‘Late Positive Component’ (LPC, linked to verbal memory) are reliably observed in healthy elderly but not in mild cognitive impairment (MCI) or AD patients [15,18,19,20]. Abnormalities in these ERP components have also been found in preAD [15], suggesting that EEG/ERP paradigms may provide sensitive biomarkers of synaptic and network alterations before any detectable cognitive dysfunction.

A limitation of traditional EEG analyses is their usual focus on pre-defined time windows, electrode locations, and frequency bands, at the expense of the overall pattern and complexity of EEG data. Methodological differences between studies also limit their scalability in large-scale studies. To address these limitations, we applied visibility graph (VG) features and machine learning to the word repetition EEG data [21]. The VG method maps one-dimensional, non-stationary time series into two-dimensional graphs based on mutual visibility between data points, allowing the exploration of the underlying dynamics of EEG data through graph-theoretical analysis [22,23]. VG is shown to preserve certain properties of the time series. For instance, periodic series yield regular graphs, while random series produce randomness [23]. Prior work demonstrates that VG is an effective approach for probing the underlying dynamics from EEG data [24], but often ignores the variable strength of network connections, which offer additional information. Therefore, we consider an improved VG method that can weigh network connections accordingly.

This paper proposes a novel analytical framework that integrates Weighted Visibility Graphs (WVG) with ensemble learning for the early detection of preAD using multichannel scalp EEG recorded during word repetition. WVG extends the natural VG approach by incorporating edge weights that reflect the visibility degree between time points, providing more correlation information of EEG dynamics [25,26]. WVG also enhances traditional ERP analysis methods by providing a more comprehensive representation of brain dynamics. Applied to the word repetition EEG data, this framework may also offer insight into how cerebral amyloidosis affects brain function in preAD.

Specifically, the framework transforms each EEG channel into a WVG, from which graph features are extracted. To reduce feature noise, *t*-tests are used for feature selection. Statistical analyses reveal structural differences in WVG networks between preAD and normal elderly participants, supporting the identification of preAD.

Next, Principal Component Analysis (PCA) reduces the input data into their principal components, which are then fed into various machine learning algorithms. To improve classification accuracy and generalization, we apply an ensemble method, which helps mitigate the tendency of individual models to overfit certain EEG trials, especially when training data are limited [27]. Our experiments demonstrate the framework’s effectiveness in distinguishing preAD from normal old participants/individuals, using both linear and non-linear classifiers.

## 2. Methods

### 2.1. Participants

Participants were recruited from the University of California, Davis (UCD) Alzheimer’s Disease Research Center (ADRC) and the UC San Diego (UCSD) Shiley-Marcos ADRC. All participants provided informed written consent in accordance with the guidelines of the UCD and UCSD Human Research Protection Programs.

Participants exhibited no significant cognitive impairment on detailed neuropsychological testing and were given a clinical diagnosis of “normal cognition” by their ADRC following a comprehensive case conference review. They were classified as preclinical AD (preAD) if they had an abnormal amyloid PET scan, indicated by increased florbetapir binding in at least two brain regions per clinical read, and met current research criteria for preclinical AD in any of its three stages [2]. Those whose amyloid PET scans were normal (no increased florbetapir binding or only mildly increased in a single brain region per clinical read) were classified as “normal old” (NO). The study included 20 patients diagnosed with preAD (mean age = 73.6 years; range: 69–81). Additionally, 20 normal old persons participated (mean age = 72.8 years; range: 64–85).

### 2.2. Word Repetition Paradigm

During each trial, participants were exposed to an auditory phrase indicating a category (e.g., “a type of wood”, “a breakfast food”), followed by the presentation of a visual target word approximately 1 s later (stimulus duration = 0.3 s, visual angle 0.4 degrees). These target words, which were nouns, had a fifty-fifty chance of being semantically congruous (e.g., ‘cedar’) or incongruous with the preceding category phrase. The congruous and incongruous words were carefully matched on usage frequency (mean = 32, SD = 48) and word length (mean = 5.8 characters, SD = 1.6).

Participants were instructed to wait for 3 s following the onset of each target word, then read/articulate the word aloud, and follow it with a yes/no judgment regarding its congruity with the preceding category. No time constraint was placed on participants’ responses. Among all category-word pairs, one-third were presented only once, one-third were presented twice, and the remaining one-third were presented three times (with congruous and incongruous pairs being counterbalanced). For items presented twice, the interval between the first and second presentations was brief (ranging from 0 to 3 intervening trials, spanning approximately 10 to 40 s). For items presented three times, the intervals between presentations were longer (ranging from 10 to 13 intervening trials, spanning approximately 100 to 140 s). The experimental data were parsed into six conditions: All New (AN), New Congruous (NC), New Incongruous (NI), All Old (AO), Old Congruous (OC), and Old Incongruous (OI) words. Further details of the experimental design have been published previously [20,21,28].

### 2.3. EEG Signal Preparation

EEG recordings were obtained across participants using 32 channels [21,28] embedded in an elastic cap (ElectroCap, Eaton OH). Electrode placements were defined by the International 10–20 system from midline (Fz, Cz, Pz, Poz), lateral frontal (F3, F4, F7, F8, FC1, FC2, FP1, FP2), temporal (T5, T6), parietal (P3, P4, CP1, CP2), and occipital sites (O1, O2, PO7, PO8). Additional sites included approximate locations of Broca’s area (Bl/Br), Wernicke’s area (Wl/Wr) and their right hemisphere homologues, and Brodmann area 41 (L41/R41). The EEG signals were sampled at 250 Hz, band-pass filtered within the range of 0.016 to 100 Hz, and offline re-referenced to averaged mastoids. Electrode impedances were kept below 5 kΩ. Data preprocessing and artifact rejection were carried out using MATLAB with the EEGLAB [29] and Fieldtrip toolboxes [30]. EEG epochs, time-locked to the onset of target words, were extracted with a duration of 2 s before and 2 s after visual word onset. Visual inspection was conducted to identify and discard non-physiological artifacts. Subsequently, independent component analysis was employed to isolate and remove eye movement artifacts.

The artifact-free EEG epochs were then extended to 8 s by mirror-padding (adding 2 s to both the beginning and end). Subsequently, they were band-pass filtered into five frequency bands (δ: 1–4 Hz, θ: 4–8 Hz, α: 8–13 Hz, β: 13–30 Hz, γ: 30–45 Hz) using zero-phase Hamming-windowed sync finite impulse response filters, as implemented in EEGLAB (pop_eegfiltnew). This function automatically determined the optimal filter order and transition bandwidth to minimize distortions and maximize time precision.

For each of the five frequency bands of interest, a high-pass filter was initially applied, followed by a low-pass filter. Transition band widths were set to be 25% of the passband edge for passband edges >4 Hz, with a −6 dB cutoff frequency at the center of the transition band. Specifically, for the 4 Hz passband, a transition bandwidth of 2 Hz was employed, while for the 1 Hz passband (δ band), a transition bandwidth of 1 Hz was utilized. Finally, both raw and band-pass filtered EEG segments were extracted, covering 1 s before and 2 s after the word onset, to facilitate further analyses.

### 2.4. Time Series Preprocessing

For each patient, we conducted 72 word repetition trials across each experimental condition. To enhance the signal-to-noise ratio in the EEG data and extract event-related information, we averaged the trials within each condition, resulting in a single averaged EEG time series per (condition, frequency band, channel) combination for each individual. Each time series was subsequently averaged into non-overlapping epochs of 80 ms, with the values of every 20 timesteps being averaged together. All-time series were uniformly shortened to cover 1 s before the stimulus onset and 2 s after it. This approach reduces signal noise and improves analysis efficiency:

Noise Reduction: This step effectively mitigated the risk of overfitting and minimized signal noise. The preprocessing technique acted as a low-pass filter, reducing the variance within individual EEG signals.

Efficiency: By reducing the signal length, we expedited the data analysis process.

### 2.5. Weighted Visibility Graphs (WVG)

The EEG signal represents the electrical activity of neurons in the brain, detected at the scalp. It exhibits prominent characteristics of non-stationarity, non-linearity, and dynamics. The VG method offers a way to explore the underlying dynamics of EEG data, converting time series into two-dimensional visual representations. Different EEG signal channels capture electrophysiological information from distinct scalp regions, enabling the creation of single-channel complex networks.1 Multiple channels yield multi-layer networks. WVG is an advanced variant of natural VG, incorporating weighted edges based on the visibility degree between corresponding data points. The construction of brain networks via WVG is illustrated schematically in Figure 1.

In constructing a WVG from univariate EEG data ${\left\{x_{i}\right\}}_{i=1}^{N}$, where $x_{i}=x\left(t_{i}\right)$, individual observations are treated as vertices. The weighted adjacency matrix W with size N×N is derived. Nodes in the WVG network correspond to time points $\left\{t_{i}\right\}$, with each edge representing a connection between two time points [31]. The nodes $x(t_{i})$ and $x(t_{j})$ are considered connected if they are “visible” from each other, which means the equation$$\frac{x\left(t_{i}\right)-x\left(t_{k}\right)}{t_{k}-t_{i}}>\frac{x\left(t_{i}\right)-x\left(t_{j}\right)}{t_{j}-t_{i}}$$
is satisfied for all time points $t_{k}$, where $t_{i}<t_{k}<t_{j}$. The absolute value of the edge weight between two nodes is then determined as follows:$$w_{i,j}=\left|\arctan\frac{x\left(t_{i}\right)-x\left(t_{j}\right)}{t_{i}-t_{j}}\right|,i<j$$

### 2.6. Feature Extraction

To capture the characteristics of the WVG networks associated with Preclinical Alzheimer’s disease (PreAD) and those of normal subjects, we compute 17 different topological features. A previous study of mild AD dementia and MCI converters introduced 12 of these features [21], including Clustering Coefficient (CC), Graph Index Complexity (GIC), Local Efficiency (LE), Global Efficiency (GE), Clustering Coefficient Sequence Similarity (CCSS), Small-worldness (SW), Size of Max Clique (SMaC), Cost of TSP (CTSP), Graph Density (GD), Independence Number (IN), Size of Minimum Cut (SMiC), Vertex Coloring Number (VCN). In this section, we introduce an additional five features: Average Weighted Degree (AWD), Degree Distribution (DD), Network Entropy (NE), Modularity (M), and Average Path Length (APL).

#### 2.6.1. Average Weighted Degree

The Average Weighted Degree is the average of the weights of all edges connected to a node. It captures the average strength of connections that each node has with its neighbors in the graph. It serves as a significant metric in discerning networks with varying topologies. This parameter is computed by averaging the weights of the links incident upon all nodes within the network [32]:$$wd=\frac{1}{N}\sum_{i\in G_{i}}s_{i}$$

#### 2.6.2. Degree Distribution Index

The degree distribution refers to the statistical distribution of node degrees across the graph. It tells us how degrees are spread out among all nodes. This metric $P_{deg}\left(k\right)$ is commonly utilized to categorize complex networks, derived by tallying the occurrence of each degree across nodes. In this study, a probability distribution entity is acquired by aligning the Poisson distribution with the degree distribution vector. The degree distribution $P_{deg}\left(k\right)$ is articulated as follows:$$P_{deg}\left(k\right)=\frac{\lambda^{k}}{k!}e^{-\lambda }$$The degree distribution index is characterized by the λ values of the fitted distribution [33].

#### 2.6.3. Network Entropy

The network entropy measures the distribution of edge weights and connectivity patterns across the graph. The computation of network entropy relies on the degree distribution.$$S=-\sum_{k}P_{deg}\left(k\right)\log P_{deg}\left(k\right)$$

#### 2.6.4. Modularity

Modularity measures the degree to which a network can be divided into distinct, non-overlapping communities or modules. It serves as a significant metric for assessing the quality of clusters, or communities, derived from network partitioning [32]. The modularity *Q* of a weighted network is defined as follows:$$Q=\frac{1}{2m}\sum_{i,j}\left(a_{i,j}-\frac{k_{i}k_{j}}{2m}\right)\delta \left(C_{i},C_{j}\right)$$Here, *m* represents the sum of weights of all links in the network, $k_{i}$ denotes the sum of weights of links attached to node *i*, $C_{i}$ indicates the community to which vertex *i* belongs, and the function $\delta (C_{i},C_{j})$ equals 1 if nodes *i* and *j* are in the same community and 0 otherwise. In our study, we applied the Louvain method [34] to allocate nodes into various communities. This method comprises two steps. Initially, each node is allocated to neighboring communities to maximize the gain in modularity *Q*. Subsequently, a new network is constructed, where each node represents a small community from the first step, and the weights of new links are determined by the sum of weights of links between nodes in the corresponding original communities. These steps are iterated until maximal modularity is achieved, and nodes cease to move. The modularity gain ΔQ is defined as follows [35]:$$\Delta Q=\left[\frac{\sum_{in}+k_{i,in}}{2m}-{\left(\frac{\sum_{tot}+k_{i}}{2m}\right)}^{2}\right]-\left[\frac{\sum_{in}□}{2m}-{\left(\frac{\sum_{tot}□}{2m}\right)}^{2}-{\left(\frac{k_{i}}{2m}\right)}^{2}\right]$$
where $\sum_{in}□$ denotes the sum of weights of links within the community *C*, $\sum_{tot}□$ represents the sum of weights of links attached to nodes in *C*, $k_{i}$ signifies the sum of weights of links attached to node *i*, $k_{i,in}$ indicates the sum of weights of links from node *i* to nodes in *C*, and *m* represents the sum of weights of all links in the network.

#### 2.6.5. Average Path Length

Average Path measures the average number of steps or connections required to travel between any two nodes in the graph. It stands as a crucial metric for gauging the information transmission capability of networks. It serves to assess the connectivity of the overall functional network, encompassing both local and distant connections. The average path length *L* is defined as follows:$$L=\frac{1}{N(N-1)}\sum_{i,j,i\neq j}l_{i,j}$$

## 3. Feature Selection

### 3.1. Two-Tailed t-Test

During the feature selection phase, our methodology employs a rigorous approach. An important consideration is that, in the literature [21,26,36], feature selection is often performed on the entire dataset. However, this approach introduces information leakage [37], because the test data have already been seen during the feature selection stage. The impact of such data leakage can be large, especially when the number of features is much larger than the number of samples. To address this issue, we randomly select 85% of the original data for training and use the remaining 15% to construct a test set. It is important to note that our feature selection process is conducted only using the training set. This step is critical because it avoids artificial inflation of classification accuracy.

The extensive feature extraction yields a total of 8676 features, which may contain considerable noise. The number of 8676 is derived from 15 channels × ((5 bands + 1 raw) × 16 single-channel features × 6 conditions) + ((5 bands + 1 raw) × 1 all-channel feature (CCSS) × 6 conditions)). To discern the statistical significance of graph features between different groups within the training data, we employ the two-tailed *t*-test in the training set to obtain significance levels (*p*-values) using the Scipy open-source library. We set a significance threshold of 0.01 and only the features that pass this test are utilized in subsequent analyses.

### 3.2. Principal Component Analysis (PCA)

Although we retain only features with a significance level of *p* < 0.01, the number of features remains large. To further reduce dimensionality, we use PCA, which linearly transforms features from a higher-dimensional space to a lower-dimensional space defined by the eigenvectors that capture the directions of greatest variance. Following the suggestion by Ahmadlou et al. [36], we employ PCA to reduce the feature space to 11 dimensions.

## 4. Machine Learning Classifiers

We evaluate various machine learning algorithms for classification, including linear logistic regression (LR), linear soft-margin support vector machines (SVM), linear discriminant analysis (LDA), k-nearest neighbor (KNN), random forest (RF), and a fully connected artificial neural network (ANN). Each algorithm is selected to serve specific purposes: logistic regression and support vector machines are chosen to evaluate linear separability. LDA models the data distributions of both classes as Gaussian distributions with equal covariances. It subsequently draws a linear decision boundary between the means of these Gaussian distributions. K-Nearest neighbor and random forest algorithms do not make assumptions about the distribution of the data. Last, the inclusion of a fully connected artificial neural network aims to approximate more intricate decision boundaries. For this purpose, we employ a simple 2-layer fully connected neural network with Rectified Linear Unit (ReLU) activations.

## 5. Ensemble

To enhance the reliability and robustness of the classifiers, we incorporate an ensemble method in conjunction with the classifiers. In our dataset, each channel comprises 72 trials, which we systematically divide into 31 distinct non-overlapping subgroups. For each of these subgroups, we independently train the classifiers, allowing them to learn from different subsets of the data. Subsequently, we employ a majority voting mechanism to aggregate the predictions generated by the individual classifiers.

This ensemble strategy leverages the diversity inherent in the training data subsets, thereby mitigating the risk of overfitting and enhancing the generalization capacity of our classification model. By combining the predictions from multiple classifiers trained on diverse data subsets, we aim for a more robust and accurate classification outcome.

## 6. Statistical Analysis on Features

As stated in Section 3.1, to address the issue of data leakage, we use a randomized approach to select 85% of the original data (40 subjects) as the training dataset. The remaining 15% serves as the testing dataset, with this process repeated for 100 rounds. Feature selection using two-tailed *t*-tests is conducted only on the training set.

The feature extraction produces a total of 8676 features across different channels, bands, and conditions. We evaluated the statistical differences between preAD patients and normal groups feature by feature. Table 1 presents the number of features selected by each frequency band and word repetition paradigm condition using the two-tailed *t*-tests, including the mean and standard deviation (std) of the 100 splits. Only features with a significance level of *p* < 0.01 are selected. We can see that the largest number of features comes from the raw band. Among the different conditions, the OI condition contributes the highest number of features.

Table 2 summarizes feature distribution after a two-tailed *t*-test across different channels, revealing potential variations in neural responses across distinct brain regions. This analysis enables us to identify channels that are particularly informative in discriminating between subject groups, thereby enhancing our understanding of the underlying neural mechanisms at play. Table 2 reveals that most selected features are derived from Fz, Pz, and Cz. This observation leads us to believe that focusing on midline sites may provide more substantial assistance in identifying preAD. The next three most helpful sites were over the temporal scalp (Wl, Wr, T6).

Moreover, Table 3 provides a summary of the most frequently selected features after the two-tailed *t*-test, offering insight into which features are particularly beneficial for distinguishing between preAD patients and normal participants in our approach. Table 3 highlights that features, such as Clustering Coefficient, Local Efficiency and Clustering Coefficient Sequence Similarity are the most selected features.

Figure 2 presents the normalized feature values (averaged across subjects) for both preAD and control groups under the OI condition, raw band, and Cz channel. Notably, the Clustering Coefficient, Local Efficiency, and CCSS values of the preAD group are lower than those of the normal group with *p* < 0.01 (marked by **). Additionally, the Average Weighted Degree, Graph Index Complexity and network entropy of the preAD group are higher than those of the normal group with *p* < 0.05, while the Average Path Length of the preAD group is lower than that of the normal group with *p* < 0.05 (marked by *). Similar distinguishable results can be observed for other conditions, bands and channel combinations.

Moreover, we visualize the separation between preAD and normal by projecting the selected features down to two dimensions using PCA, based on one representative split out of 100, as shown in Figure 3. All classifiers are able to separate the two groups, even in two dimensions, although there are a small number of misclassified points.

The ten most important features for each two-dimensional PCA projection from Figure 3 are listed in Table 4. The raw and delta bands produced the largest number of features, with the most common being Clustering Coefficient and Local Efficiency in electrodes Fz, Pz, and Cz, as well as Clustering Coefficient Sequence Similarity across all channels.

## 7. Machine Learning Results

Each model undergoes training and testing repeatedly for 100 splits, where the dataset is randomly split into a training set comprising 85% of the subjects and a test set containing the remaining 15%, with a matched number of preAD and normal subjects. These repeated evaluations validate the framework’s generalizability to new, unseen patients—supporting its potential for real-world applications. Classification metrics, in terms of accuracy, precision, recall, and AUROC, are computed based on the performance on the test set, with the results averaged across all 100 splits for each model. Moreover, we also present the K-fold (K = 8) cross-validation results using the same data and features in the Appendix A.

### 7.1. Classification Without Ensemble

The discrimination between preAD and Normal individuals averaged 88% across all classifiers, as shown in Table 5. The K-Nearest Neighbor, SVM, and Random Forest classifier achieved a discrimination accuracy > 89% for preAD vs Normal. In comparison, if we use a standard VG without ensemble methods, we only achieved an accuracy of 79%. Precision, recall, and AUROC are also reported in Table 5. We note that the std is relatively high. This is due to the inherent issue that the number of samples is relatively small. Any other algorithms with similar average performance would have the same range of std.

### 7.2. Classification with Ensemble

To enhance the reliability and precision of our classification, we incorporate an ensemble method in conjunction with the classifiers. We partition the trials in each channel into 31 distinct non-overlapping subgroups. For each subgroup, classifiers are trained independently, enabling them to learn from diverse data subsets. Subsequently, we employ a majority voting mechanism to consolidate predictions from individual classifiers. We observe a consistent improvement in accuracy across all classifiers (Table 6). This ensemble approach yields an accuracy of approximately 91%. Additionally, the Random Forest achieved the highest discrimination accuracy of 92%.

### 7.3. Reduce Channel/Trials/Band Without Ensemble

In our dataset, we observe 15 distinct channels, each comprising 72 trials. To improve clinical efficiency, we reduce the number of channels to expedite electrode setup and minimize the number of trials per channel to reduce data collection time.

Specifically, we use the training set to select five channels (Fz, Pz, Cz, Wl, and Wr), which yield the most features after the two-tailed *t*-test. The classification results with these five reduced channels remain largely unchanged (Table 7). Moreover, reducing the number of trials per channel to 30 results in accuracy that remains largely stable or experiences a slight decrease (Table 8).

Similarly, we can also reduce the number of bands used to make the classification while maintaining similar classification accuracy (Table 9). Although reducing the number of bands does not necessarily reduce EEG data collection time, it can reduce the number of features and computational costs.

### 7.4. Reduce Channel/Trials/Band with Ensemble

When we integrate ensemble techniques with the above data reduction strategies, we observe an overall accuracy increase of approximately 2–3% across all scenarios (Table 10, Table 11 and Table 12). This suggests that leveraging ensemble methods can effectively complement data reduction efforts, enhancing the robustness and performance of our classification models.

## 8. Discussion

The primary contribution of this paper lies in the development and validation of a novel analytical framework for the early detection of preclinical Alzheimer’s disease using cognitive ERP/EEG. The integration of Weighted Visibility Graphs (WVG) with ensemble learning techniques offers a robust approach to identify preAD participants with classification accuracy up to 92%. This degree of classification accuracy is comparable to AD biomarker platforms recently approved by the FDA to identify patients with amyloid pathology associated with AD [38]. Also remarkable is that this high degree of accuracy was achieved in a sample of preclinical AD, who have increased amyloid binding on florbetapir PET scans, but no significant cognitive deficits were evident on comprehensive neuropsychological testing conducted in an ADRC setting.

Specifically, this paper makes the following contributions:Integration of WVG with Ensemble Learning: Our framework of integrating WVG and ensemble Learning enhances traditional ERP and EEG analysis methods for the early detection of AD in its preclinical stages.Experimental Validation: The efficacy of the proposed framework was demonstrated through experimentation on a dataset comprising 20 preAD and 20 normal old. The results showed that the framework achieves an accuracy up to 92% with both linear and non-linear classifiers, highlighting its potential clinical utility. Some specific strengths of our analytic approach are highlighted below in Section 8.1 and Section 8.2.Improving clinical efficiency: Our experimental results demonstrate that our framework can achieve comparable classification results while utilizing less data, e.g., by employing fewer task conditions, a reduced number of channels and filter bands, and a smaller number of trials per channel. These outcomes indicate the potential of saving valuable clinical time.

An important study limitation is the modest size of our preclinical AD sample (*n* = 20). Therefore, the replication of these results in larger and independent samples is essential and will be a focus of our future research. Another limitation is that we did not obtain tau PET or tau biomarkers from CSF or plasma in the majority of these participants.

### 8.1. Ensemble

Ensemble learning is a powerful technique in machine learning where multiple models are combined to improve overall performance and robustness. By combining the predictions of multiple models, ensemble methods often achieve better accuracy than any individual model. This is because the errors of individual models can cancel each other out. Ensembles can also reduce the variance of the model. Moreover, ensemble models are generally more robust to noise and outliers in the data. The combined decision-making process helps in smoothing out irregularities and makes the model less sensitive to the peculiarities of the training data.

### 8.2. Classification Significance of Conditions/Bands/Channels/Features

Among the different conditions analyzed, the OI condition (Old incongruous words) contributes the highest number of features, as shown in Table 1. This finding suggests that the OI condition may be particularly sensitive to neural changes associated with preAD. Moreover, we can see that the largest number of features comes from the raw band. This indicates that raw EEG signals hold significant information beyond that obtained within any single traditional EEG frequency band (e.g., α, β), which is critical for distinguishing between preAD and normal subjects.

The majority of selected features are derived from midline sites, specifically Fz, Pz, and Cz, as shown in Table 2. These midline channels are known to be sensitive to changes in preAD relative to robust normal elderly on the ERP word repetition and congruity effects [15]. For example, the preAD group showed a severe reduction in the size of the P600 repetition effect, which was largest over the centro-parietal midline channels in robust normal elderly [15]. The same group of preAD also showed a reduction in the typical centro-posterior N400 effect; their N400 effect was largest over centro-anterior scalp sites. The midline sites are known to be involved in a variety of cognitive functions, including attention, executive function, and memory processing [39]. Midline/medial brain regions such as the posterior cingulate and precuneus are among the earliest brain predilection sites for early amyloid deposition in AD [40]. Our observations may help future research on EEG source localization techniques—which, due to poor spatial resolution, struggle to accurately pinpoint the precise location of brain activity [41]—in determining whether these brain regions generate the midline cognitive ERP effects. Nevertheless, focusing on these midline sites may provide more substantial assistance in identifying preclinical AD. It is possible that changes in the connectivity and activity patterns in these regions are more pronounced or detectable, making them reliable indicators for early diagnosis.

Following the midline channels, we found that left and right temporal channels (Wl, Wr, T6, T5) provided the next largest number of features used in discriminating PreAD from normal old (Table 2). This may reflect that the temporal cortex is a predilection site for neurofibrillary tangles in early AD (Braak stage II–III). Also, the temporal cortex is particularly critical for semantic processing and the classification task used in our ERP experiment.

Table 3 and Figure 2 highlight that features, such as Clustering Coefficient, Local Efficiency, and Clustering Coefficient Sequence Similarity are the most selected features. In graph theory, the clustering coefficient of a node measures the extent to which its neighbors form a complete graph (i.e., how interconnected the neighbors are). In the context of visibility graphs constructed from EEG signals, a high clustering coefficient indicates that the neighboring time points (nodes) have strong mutual visibility, reflecting a robust local network structure. AD is characterized by progressive neural degeneration, which disrupts both local and global brain connectivity [42]. This disruption can manifest as changes in the clustering coefficient within EEG-derived visibility graphs. A decrease in the clustering coefficient in EEG visibility graphs could indicate a decrease in local synaptic density, the loss of neuronal connectivity and synaptic dysfunction, all of which are early signs of AD. Because synaptic density is one of the strongest predictors of AD severity [10], tracking changes in the clustering coefficient over time may allow clinicians to sensitively monitor AD progression.

Local efficiency of a node in a graph measures the efficiency of information transfer within its immediate neighborhood. It is calculated as the average efficiency of the subnetwork formed by the node’s neighbors, excluding the node itself. High local efficiency indicates that the neighbors are well-connected, facilitating efficient local information processing. A decrease in local efficiency in EEG visibility graphs can indicate early disruptions in local neural circuits. This can be an early biomarker for AD, as synaptic dysfunction and local network breakdown are early pathological features of the disease.

Clustering Coefficient Sequence Similarity (CCSS) measures the resemblance between the clustering coefficient sequences of nodes across visibility graphs (VGs) derived from different EEG time series channels. AD disrupts normal brain network organization, leading to alterations in local clustering properties across different brain regions. This can result in reduced CCSS, as the similarity in local network organization between different regions becomes less pronounced. A decrease in CCSS between EEG channels could indicate early disruptions in brain connectivity, especially in neural networks associated with AD pathology.

## 9. Conclusions

In summary, we report here an effective framework for early detection of preclinical Alzheimer’s disease using multichannel EEG signals by combining Weighted Visibility Graphs, statistical feature selection, PCA-based dimensionality reduction, and ensemble machine learning. The optimization of these models achieved up to a 92% classification accuracy in discriminating preclinical AD from normal old, demonstrating its potential as a robust diagnostic tool with possible clinical utility. These findings provide further evidence that EEG/ERP measures may be sensitive to preclinical disease and that synaptic dysfunction is very common in preclinical AD, and could be the earliest type of pathophysiology [3].

An important study limitation is our modest sample size. Therefore, the replication of these results in larger and independent samples is essential before deploying them in clinical settings or in clinical trials for AD prevention. A focus of our future research will be to study larger samples of preclinical AD, ideally across multiple ADRC sites and in clinical settings such as subjective memory complaints or MCI. It will also be important to follow up on the long-term clinical outcomes and neuropathological diagnoses of these well-characterized ADRC participants. We will be able to test if the misclassified preAD cases are less likely to convert to MCI or show subsequent decline in memory compared with preAD cases with abnormalities detected by our WVG and ensemble learning method. Prognostic markers are extremely valuable in preclinical AD, as they can identify who is most in need of an intervention (pharmacologic, behavioral, lifestyle), some of which may be invasive or carry substantial risks (e.g., brain hemorrhage or edema associated with amyloid immunotherapies [43]).

## Figures and Tables

**Figure 1 bioengineering-12-00814-f001:**
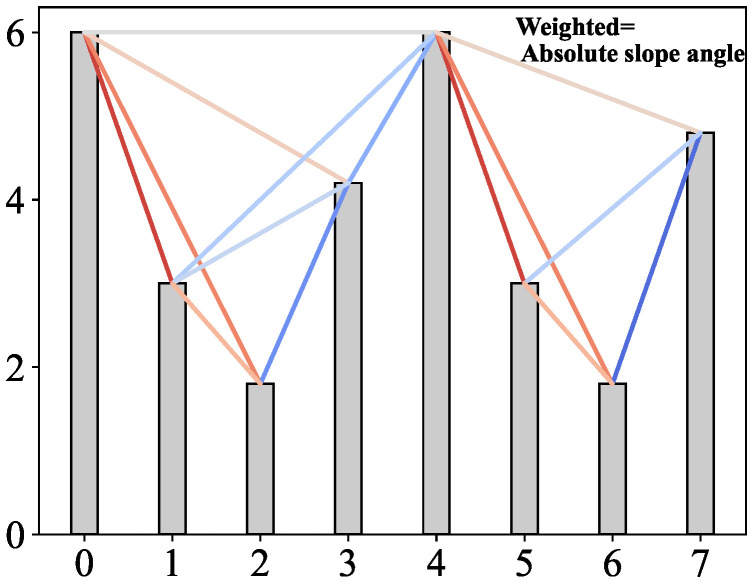
Example of a time series (8 data points) and the associated graph derived from the weighted visibility algorithm. In the graph, every node corresponds, in the same order, to time series data. Every edge is weighted by the slope angle.

**Figure 2 bioengineering-12-00814-f002:**
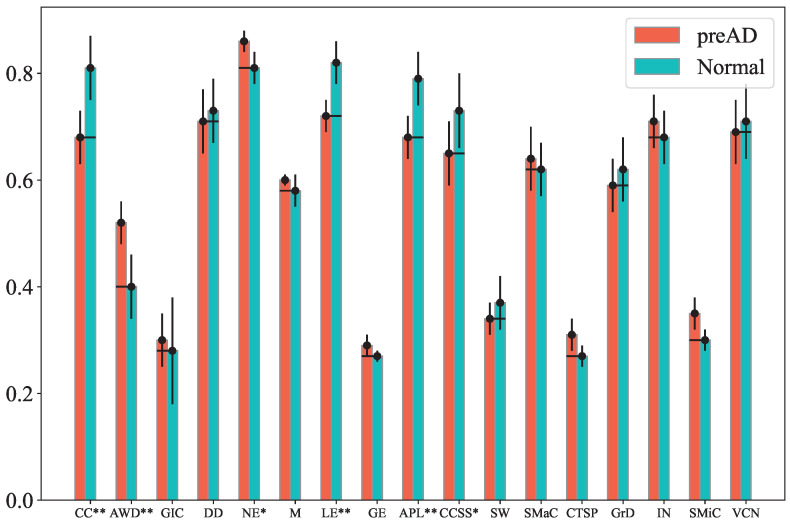
The features (averaged across participants) are for both preAD and normal groups, with error bars representing the standard error across subjects. The level of significance is calculated using a two-tailed *t*-test across all subjects. ** denotes a significant level (*p* ≤ 0.01), while * indicates (*p* ≤ 0.05).

**Figure 3 bioengineering-12-00814-f003:**
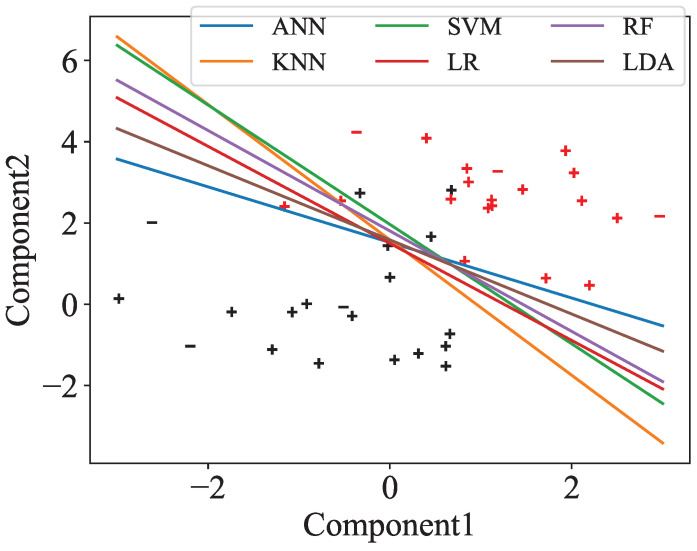
Two-dimensional PCA projections of the data with associated decision boundaries for all classifiers and data points, based on one representative split out of 100. We use two colors (red and black) to represent the two participant groups, and two markers (‘+’ and ‘−’) to distinguish between the training and test sets. For each plot, the PCA components were computed using only the data in that specific plot to reflect the actual input to the ML algorithms. In two dimensions, it is evident that preAD and normal groups are linearly separable with the features we extracted.

**Table 1 bioengineering-12-00814-t001:** The number of features (mean-std across 100 splits) produced by each band after two-tailed *t*-test.

	NA	NC	NI	OA	OC	OI	Total
Raw	4.1~1.7	4.1~3.5	2.2~2.5	9.4~3.7	6.5~3.1	20.1~4.3	**46.4~15.3**
Delta	0.9~0.3	4.0~1.9	0.2~0.1	3.9~2.6	2.9~1.8	8.4~3.9	20.3~13.6
Theta	2.1~1.1	1.9~0.7	0.6~0.2	0.5~0.3	1.6~0.5	0.6~0.3	7.3~3.7
Alpha	2.2~1.6	2.3~1.7	1.3~0.9	1.1~1.0	1.1~0.8	0.9~0.6	8.9~2.7
Beta	0.4~0.2	1.3~0.4	0.6~0.4	0.3~0.2	0.4~0.3	2.3~2.0	5.3~3.5
Gamma	0.5~0.5	3.7~2.6	0.4~4.6	5.0~4.6	2.7~4.6	3.0~0.3	15.3~4.3
Total	10.2~1.4	17.3~4.2	5.3~1.8	20.2~7.8	15.2~6.3	**35.3~8.6**	**103.5~29.5**

**Table 2 bioengineering-12-00814-t002:** The number of features (mean-std across 100 splits) produced by each channel after a two-tailed *t*-test. e_c is the CCSS feature across all channels.

	NA	NC	NI	OA	OC	OI	Total
Fz	0.9~0.7	0.4~0.3	0.5~4.5	1.7~0.4	1.9~0.4	4.9~3.5	**10.3~6.4**
Pz	0.2~0.2	2.1~1.5	0.3~0.2	2.3~1.2	1.9~1.3	8.6~1.8	**15.4~5.7**
Cz	0.5~0.3	2.8~1.9	0.6~0.8	3.6~1.9	3.3~3.2	6.6~1.7	**17.4~3.5**
F7	0.7~0.3	1.0~0.6	0.5~0.4	0.6~0.2	1.0~0.6	0.6~0.5	4.4~2.4
F8	0.4~0.2	0.5~0.3	0.3~0.2	0.7~0.3	0.4~0.2	0.1~0.1	2.4~0.6
Bl	0.1~0.1	0.7~0.3	0.1~0.1	0.1~0.1	0.1~0.1	0.2~0.1	1.3~0.4
Br	0.2~0.1	0.8~0.6	0.1~0.1	0.0~0.0	0.1~0.1	0.2~0.1	1.4~0.4
L41	1.6~1.5	0.4~0.5	0.1~0.2	1.2~0.9	0.1~0.1	1.1~0.9	4.5~3.0
R41	0.3~0.5	0.6~0.4	0.4~0.6	0.3~0.4	0.0~0.0	1.8~0.9	3.4~1.4
Wl	1.3~0.8	0.5~0.4	0.5~0.7	1.0~0.8	0.9~0.6	4.3~3.2	8.5~4.2
Wr	0.7~0.2	0.3~0.5	0.6~0.9	1.9~1.1	1.2~0.7	2.8~1.6	7.5~1.9
T5	0.9~0.7	0.7~0.6	0.4~0.5	0.8~0.7	0.9~0.7	1.2~1.3	4.9~4.6
T6	0.4~0.6	0.8~0.6	0.3~0.6	4.0~2.3	1.7~0.5	0.3~0.2	7.5~3.7
O1	0.2~0.3	0.1~0.1	0.3~0.2	0.2~0.1	0.0~0.0	0.5~0.4	1.3~1.4
O2	0.5~1.0	0.9~0.8	0.0~0.0	1.1~1.2	0.6~0.8	0.3~0.4	3.4~2.6
e_c	1.3~1.3	4.7~1.0	0.3~1.9	0.7~3.0	1.1~1.0	1.8~1.6	9.9~2.3
Total	10.2~1.4	17.3~4.2	5.3~1.8	20.2~7.8	15.2~6.3	**35.3~8.6**	**103.5~29.5**

**Table 3 bioengineering-12-00814-t003:** The number of features (mean-std across 100 splits) produced after two-tailed *t*-test under different conditions.

	NA	NC	NI	OA	OC	OI	Total
CC	1.3~2.6	0.4~1.4	0.1~0.1	4.2~1.3	2.4~1.5	7.8~2.5	**16.2~4.3**
AWD	0.8~0.5	1.2~0.8	0.4~0.5	1.4~0.9	1.9~0.6	3.7~2.1	9.4~1.4
GIC	0.8~0.8	0.9~0.6	0.5~0.3	1.4~0.7	0.9~0.3	2.9~1.2	7.4~1.3
DD	0.9~0.7	0.5~0.4	0.8~0.6	0.4~0.2	0.7~0.4	1.0~0.8	4.3~5.5
NE	0.8~0.6	0.7~0.6	0.4~1.1	1.2~0.5	1.3~0.7	3.0~1.7	7.4~4.8
M	0.7~0.6	1.1~0.8	0.3~0.2	0.8~0.6	0.5~0.3	1.1~0.9	4.5~1.3
LE	0.6~2.1	2.3~1.1	0.5~0.9	3.8~2.1	2.4~2.1	5.6~2.1	**15.2~4.4**
GE	0.2~0.1	0.4~0.3	0.6~0.3	0.2~0.1	0.0~0.0	1.0~0.4	2.4~1.4
APL	0.5~0.4	1.3~0.2	0.1~0.1	2.5~2.1	1.1~0.9	2.8~2.1	8.3~3.5
CCSS	0.9~1.7	6.2~3.2	0.0~0.0	1.7~0.6	1.6~1.3	3.0~1.2	**13.4~3.5**
SW	0.6~0.3	0.2~0.2	0.1~0.1	0.0~0.0	0.2~0.1	0.3~0.2	1.4~3.5
SMaC	0.4~0.3	0.6~0.3	0.2~0.2	0.3~0.2	0.5~0.3	0.4~0.2	2.4~2.6
CTSP	0.3~0.7	0.3~0.2	0.3~0.2	0.7~0.5	0.6~0.4	1.0~0.3	3.2~5.3
GD	0.4~0.4	0.5~0.4s	0.2~0.1	0.3~0.2	0.6~0.3	0.7~0.4	2.7~1.4
in	0.2~0.2	0.1~0.1	0.3~0.2	0.5~0.3	0.0~0.0	0.3~0.2	1.4~0.5
SMiC	0.3~0.2	0.4~0.3	0.2~0.1	0.2~0.2	0.1~0.2	0.3~0.2	1.5~0.5
VCN	0.5~0.4	0.2~0.2	0.3~0.2	0.6~0.3	0.4~0.2	0.4~0.2	2.4~1.5
Total	10.2~1.4	17.3~4.2	5.3~1.8	20.2~7.8	15.2~6.3	**35.3~8.6**	**103.5~29.5**

**Table 4 bioengineering-12-00814-t004:** Top 10 features for each two-dimensional PCA projection from Figure 3.

preAD vs. Normal	Band/Electrode/Feature	Magnitude
Component 1	Raw/Cz/CC	0.245
	Raw/Cz/LE	0.245
	Raw/Pz/CC	0.244
	Delta/Pz/CC	0.244
	Delta/CCSS	0.232
	Raw/Fz/LE	0.223
	Raw/Fz/AWD	0.213
	Gamma/CCSS	0.210
	Raw/Fz/GIC	0.204
	Gamma/Cz/CC	0.198
Component 2	Raw/Cz/AWD	0.301
	Raw/Cz/GIC	0.301
	Raw/Fz/CC	0.298
	Raw/Fz/NE	0.297
	Delta/Fz/CC	0.272
	Raw/CCSS	0.240
	Delta/Wl/LE	0.239
	Gamma/Cz/CC	0.214
	Delta/Wl/AWD	0.205
	Raw/Wr/NE	0.176

**Table 5 bioengineering-12-00814-t005:** Classification without ensemble (mean-std across 100 splits).

	Accuracy	Precision	Recall	AUROC
Neural Network	87.45~13.67	92.33~14.22	80.34~25.34	87.45~16.34
3-Nearest Neighbor	89.45~13.56	92.33~14.22	84.23~21.45	85.45~18.34
SVM	89.45~15.34	92.33~14.22	84.34~22.45	86.34~18.34
Logistic Regression	87.34~14.23	93.89~13.93	81.34~23.45	85.34~19.34
Random Forest	89.34~14.34	94.34~12.44	85.34~21.34	89.32~18.34
LDA	87.34~14.45	91.34~17.56	82.34~21.45	85.23~18.34

**Table 6 bioengineering-12-00814-t006:** Classification with ensemble (mean-std across 100 splits).

	Accuracy	Precision	Recall	AUROC
Neural Network	91.00~14.34	92.33~19.06	85.00~25.00	84.75~19.00
3-Nearest Neighbor	91.45~14.45	93.83~16.10	85.00~23.98	82.00~16.99
SVM	91.01~14.45	89.33~20.62	84.50~25.19	84.00~20.47
Logistic Regression	91.34~14.34	91.83~19.51	84.50~25.19	84.25~19.89
Random Forest	92.01~13.56	90.33~21.50	84.00~26.34	84.88~28.09
LDA	89.34~14.56	91.17~18.02	87.00~23.04	84.75~24.48

**Table 7 bioengineering-12-00814-t007:** Reduce number of channels to 5 without ensemble (mean-std across 100 splits).

	Accuracy	Precision	Recall	AUROC
Neural Network	88.45~13.45	96.34~11.34	81.34~23.34	91.34~16.34
3-Nearest Neighbor	88.34~14.45	94.23~14.34	81.34~22.45	84.34~15.23
SVM	86.3~15.34	91.34~14.34	82.34~21.34	87.34~15.34
Logistic Regression	88.34~12.56	95.34~13.44	82.34~22.34	91.34~14.45
Random Forest	89.34~14.78	91.34~15.34	85.34~22.45	86.34~22.12
LDA	87.45~14.46	94.34~14.34	84.34~22.45	87.34~16.34

**Table 8 bioengineering-12-00814-t008:** Reduce number of trials to 30 without ensemble (mean-std across 100 splits).

	Accuracy	Precision	Recall	AUROC
Neural Network	87.34~14.45	81.34~33.45	78.34~32.34	85.34~21.34
3-Nearest Neighbor	88.34~14.65	84.23~26.45	81.34~27.34	82.34~20.34
SVM	86.34~14.45	81.23~32.34	81.45~31.34	84.23~23.34
Logistic Regression	86.34~14.34	81.34~32.45	78.34~31.34	84.45~21.34
Random Forest	87.45~16.45	79.34~31.56	81.45~31.45	82.34~23.65
LDA	86.34~14.34	83.45~32.45	72.34~30.34	81.34~23.45

**Table 9 bioengineering-12-00814-t009:** Reduce number of bands to 3 (Raw, Delta, Gamma) without ensemble (mean-std across 100 splits).

	Accuracy	Precision	Recall	AUROC
Neural Network	87.34~13.45	94.45~14.45	84.34~23.34	83.34~22.34
3-Nearest Neighbor	88.34~14.34	94.45~14.45	87.34~20.45	81.34~21.45
SVM	87.34~16.34	94.45~14.45	91.34~22.34	82.34~22.45
Logistic Regression	87.34~15.34	94.45~14.45	91.34~22.34	84.34~22.45
Random Forest	89.56~14.56	93.34~13.45	90.34~17.34	81.34~21.45
LDA	88.34~14.45	94.45~14.45	90.34~17.34	81.23~21.34

**Table 10 bioengineering-12-00814-t010:** Reduce number of channels to 5 with ensemble (mean-std across 100 splits).

	Accuracy	Precision	Recall	AUROC
Neural Network	90.34~15.45	91.34~15.34	90.34~21.45	89.34~21.34
3-Nearest Neighbor	90.34~15.23	91.34~15.34	90.56~20.56	81.45~20.46
SVM	90.45~14.45	92.34~14.45	91.56~20.56	86.45~20.54
Logistic Regression	90.34~15.45	91.45~16.34	91.33~21.56	88.43~20.45
Random Forest	91.34~14.34	91.56~15.39	90.34~20.45	89.45~16.23
LDA	91.34~15.45	90.34~15.56	90.45~21.45	85.34~18.34

**Table 11 bioengineering-12-00814-t011:** Reduce number of trials to 30 with ensemble (mean-std across 100 splits).

	Accuracy	Precision	Recall	AUROC
Neural Network	90.34~14.46	90.34~19.34	90.34~23.45	82.34~21.34
3-Nearest Neighbor	91.35~15.45	90.34~22.34	92.34~22.45	83.34~16.34
SVM	90.45~14.64	90.34~18.34	90.34~22.34	81.34~20.34
Logistic Regression	91.45~15.56	90.45~17.34	92.34~22.90	81.34~20.34
Random Forest	89.45~14.45	88.34~21.45	90.34~22.34	81.45~23.45
LDA	91.34~14.45	89.34~22.34	89.34~22.34	82.34~23.45

**Table 12 bioengineering-12-00814-t012:** Reduce number of bands to 3 with ensemble (mean-std across 100 splits).

	Accuracy	Precision	Recall	AUROC
Neural Network	91.45~15.45	90.34~14.34	90.45~22.45	86.34~12.45
3-Nearest Neighbor	91.45~15.45	94.34~15.34	88.34~20.34	85.23~12.45
SVM	90.34~15.45	95.23~15.45	90.23~18.34	84.45~17.45
Logistic Regression	92.45~15.45	95.23~15.45	89.34~21.34	88.34~18.34
Random Forest	91.45~14.45	93.45~16.45	90.34~20.34	87.34~15.34
LDA	90.34~17.34	90.23~19.45	90.34~20.34	86.34~21.64

## Data Availability

The datasets used in this study were provided with permission from the Alzheimer’s Disease Research Centers at the University of California, San Diego, and the University of California, Davis. Requests for access to these datasets should be directed to John Olichney, M.D.

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
