# Peer review of "Unveiling Early Signs of Preclinical Alzheimer’s Disease Through ERP Analysis with Weighted Visibility Graphs and Ensemble Learning"

_bioengineering, 2025, doi:10.3390/bioengineering12080814_

Round 1
Reviewer 1 Report
Comments and Suggestions for Authors
The manuscript is devoted in essence to the classification of EEG data obtained during specific task. The goal of the classifier is the detection of pre-clinical signs of Alzheimer’s disease. As the authors note, early detection of Alzheimer’s disease is important for effective therapeutic interventions and optimized enrollment for clinical trials.
Experimental approach is based on previous works by the same team where multichannel scalp EEG signals are recorded during word repetition. In this paradigm, designed to elicit brain activity related to language and memory processing, the laboratory has identified several patterns that are promising for identification of various stages of Alzheimer’s disease. This paper proposes a novel analytical framework that integrates relatively novel approach called Weighted Visibility Graphs and modern machine learning techniques.
Major points:
The test set is very small (only 6 subjects). However, the authors claim “identifying very early AD from EEG signals”. The authors should comment on possible future work on increasing the number of participants. Could the framework be applied as is for the analysis of new patients or should it be trained further before real applications?
Equation at the bottom of p. 5:
Probably there should be t_j – t_i in the denominator on the right-hand side.
Derivatives are very susceptible to noise. It means that graph architecture would largerly depend on noise. Could the authors comment on this?
“The absolute value of the edge weight between two nodes” are then expressed in the next equation through the arctan function, which might have negative values. How the absolute value can be negative?
Figures are of bad quality and should be made readable.
Section 8 (discussion) has the typical structure of AI generated list. I recommend reshaping the structure to make it more like human-written.
Author Response
We sincerely thank you for the time and effort spent reviewing our manuscript. We greatly appreciate the constructive feedback and insightful suggestions, which have helped us improve the quality and clarity of our work. We have provided a detailed, point-by-point response to each reviewer's comment, indicating the corresponding changes made in the manuscript. Revisions are highlighted in blue with wavy underlines, and key emphasized text is marked in red.
Comments 1: The test set is very small (only 6 subjects). However, the authors claim “identifying very early AD from EEG signals”.
Response 1: Thank you for pointing this out. We would like to clarify that our study includes a total of 40 subjects, as stated in the last sentence of Section 2.1 (page 3): “The study included 20 patients diagnosed with preAD (mean age = 73.6 years; range: 69–81). Additionally, 20 cognitively normal older adults participated (mean age = 72.8 years; range: 64–85).” We have marked this sentence in red in the revised manuscript for better visibility.
While the reviewer is correct that each iteration used a test set of n=6 (15% randomly selected), this was repeated 100 times and therefore all 40 subjects contributed to the study results.
Comments 2: The authors should comment on possible future work on increasing the number of participants.
Response 2: Thank you for your valuable suggestion. We have added a discussion of limitations and potential future work regarding increasing the number of participants in the Discussion section (Section 8, page 15, second paragraph) and the Conclusion section (Section 9, page 16, last paragraph). These additions are highlighted in blue with wavy underlines for easy reference.
Comments 3: Could the framework be applied as is for the analysis of new patients or should it be trained further before real applications?
Response 3: Yes, once our framework is trained, it can be applied as is to analyze new patients without requiring further retraining. As described in Section 3.1, Section 6 (first sentence), and Section 7 (first sentence)—marked in red in the revised manuscript—we randomly split the dataset (20 preAD and 20 normal subjects) into a training set comprising 85% of the participants (17 preAD and 17 normal) and a test set comprising the remaining 15% (3 preAD and 3 normal). This setup simulates applying the trained model to new, unseen patients.
To ensure robustness and reduce variance, we repeated the training and testing procedure 100 times with different random splits. This repeated evaluation demonstrates the model’s generalizability and supports its applicability to new patients in real-world scenarios. We have emphasized this point in the first paragraph of Section 7 (page 12) and highlighted it in blue with a wavy underline in the revised manuscript.
In the future, we plan to also apply our trained model to new independent samples of preclinical AD, and to study larger samples.
Comments 4: Equation at the bottom of p. 5: Probably there should be t_j – t_i in the denominator on the right-hand side.
Response 4: Thank you for catching this typo. We have corrected the equation by updating the denominator on the right-hand side to t_j – t_i. The revised equation is now highlighted in blue with a wavy underline in the updated manuscript (section 2.5, page 5, last line).
Comments 5: Derivatives are very susceptible to noise. It means that graph architecture would largely depend on noise. Could the authors comment on this?
Response 5: We agree that derivatives are indeed sensitive to noise, which could affect the structure of the weighted visibility graph. To mitigate this issue, we apply preprocessing steps to minimize signal noise (e.g. averaging the EEG across numerous trials of similar type, rejecting trials with non-physiological artifacts) before constructing the graph. This reduces the vulnerability of the resulting graph architecture to noise.
The specific preprocessing techniques used to denoise the EEG signals are described in Section 2.4 of the manuscript, which we have marked in red in the revised version page 5. These steps ensure that the visibility graph reflects meaningful structural patterns rather than noise.
Comments 6: “The absolute value of the edge weight between two nodes” are then expressed in the next equation through the arctan function, which might have negative values. How the absolute value can be negative?
Response 6: Thank you for pointing this out. To clarify, we use the absolute value of the arctan function output as the edge weight to ensure all weights are non-negative. We have revised the equation accordingly by explicitly adding the absolute value notation around the arctan function and highlighted the correction in blue with a wavy underline in the revised manuscript (section 2.5, page 6).
Comments 7: Figures are of bad quality and should be made readable.
Response 7: Thank you for your feedback. We have improved the readability and visual quality of all figures in the manuscript. Enhancements include increasing the resolution, enlarging fonts, and adjusting layout elements to ensure clarity.
Comments 8: Section 8 (discussion) has the typical structure of AI generated list. I recommend reshaping the structure to make it more like human-written.
Response 8: Thank you for your suggestion. We have revised the Discussion section to improve its flow and coherence. We expanded the section by elaborating on the practical implications of our findings and by outlining potential limitations and directions for future work (Section 8). Additionally, we added a new Conclusion section (Section 9, page 16) to summarize the key contributions and insights of our study. These additions are highlighted in blue with wavy underlines for easy reference.
Reviewer 2 Report
Comments and Suggestions for Authors
The article require the major revision to address several issues:
- The article does not have conclusions part at all. The Discussion part is very small and covers only the basic raw data processing results.
- Figures quality is very low, figures are nearly unreadable.
- The 2D PCA requires the explained variance computation.
- Figure 3 is drawn as component1 vs component1 graph.
The overall scientific design of the paper is somewhat flawed, it describes a basic machine learning model for processing a limited set of EEG data. There are no validation tests, most of the paper describes the raw machine learning procedure as is, without connecting the obtained results to real problems and with showing all internals - like jupyter notebook (without validation part) converted into the article. I would suggest to extend at least the discussion part of article and describe some practical implications of the obtained results. The approach has some novelty in it, but it got lost in the long tedious tables of nearly same results of similar machine learning models working on the limited dataset that perform with nearly equal quality metrics.
Author Response
We sincerely thank you for the time and effort spent reviewing our manuscript. We greatly appreciate the constructive feedback and insightful suggestions, which have helped us improve the quality and clarity of our work. We have provided a detailed, point-by-point response to each reviewer's comment, indicating the corresponding changes made in the manuscript. Revisions are highlighted in blue with wavy underlines, and key emphasized text is marked in red.
Comments 1: The article does not have conclusions part at all. The Discussion part is very small and covers only the basic raw data processing results.
Response 1: Thank you for your valuable feedback. While a conclusion section is optional, we agree that including one would strengthen the structure and clarity of the manuscript. Based on your suggestion, we expanded the section by elaborating on the practical implications of our findings and by outlining potential limitations and directions for future work (Section 8). Additionally, we added a new Conclusion section (Section 9, page 16) to summarize the key contributions and insights of our study. These additions are highlighted in blue with wavy underlines for easy reference.
Comments 2: Figures quality is very low, figures are nearly unreadable.
Response 2: Thank you for your comment. We have improved the readability and visual quality of all figures in the manuscript. Enhancements include increasing the resolution, enlarging fonts, and adjusting layout elements to ensure clarity.
Comments 3: The 2D PCA requires the explained variance computation.
Response 3: Thank you for the comment. We acknowledge the importance of explained variance in results. In our experiment, the dataset is randomly split into a training set comprising 85% of the subjects and a test set containing the remaining 15%, as described in Section 3.1, Section 6 (first sentence), and Section 7 (first sentence) — marked in red in the revised manuscript. This process is repeated for 100 rounds to account for variability and improve robustness.
Since each split results in a different selection of training and test subjects, the resulting decision boundaries also vary. Plotting all decision boundaries from all 100 splits would make the figure unreadable. To avoid visual clutter and provide a clear visual representation, the 2D PCA figure shows only one representative example of such a split. In the revised manuscript, we have clarified this in the figure caption and description (page 11, highlighted in blue with wavy underlines). In the revised figure, we also use different markers to distinguish the training and test subject.
Comments 4: Figure 3 is drawn as component1 vs component1 graph
Response 4: Thank you for pointing out this labeling error. We have corrected the axis labels in Figure 3 (page 11) to accurately reflect Component 1 vs. Component 2, as intended.
Comments 5: There are no validation tests, most of the paper describes the raw machine learning procedure as is, without connecting the obtained results to real problems and with showing all internals - like jupyter notebook (without validation part) converted into the article. I would suggest to extend at least the discussion part of article and describe some practical implications of the obtained results.
Response 5: Thank you for your constructive feedback. We would like to clarify that validation has been performed through a repeated hold-out testing procedure. Specifically, we randomly split the dataset into a training set comprising 85% of the subjects and a test set containing the remaining 15%. This process was repeated 100 times to ensure robustness and reduce variance. These repeated evaluations validate the framework’s generalizability to new, unseen patients—supporting its potential for real-world applications. We have emphasized this point in the first paragraph of Section 7 (page 12) and highlighted it in bule with a wavy underline in the revised manuscript.
To improve the clarity and structure of the paper, we have moved several parts of the results section to the appendix (e.g. Tables A3-A4 on page 18 marked in red). Additionally, we have expanded the Discussion section (Section 8) and added a Conclusion section (Section 9, page 16) to better articulate the practical implications of the obtained results, including how the proposed framework could assist in the early detection and monitoring of preclinical Alzheimer’s disease (preAD), as well as to address potential limitations and outline future directions for clinical deployment.
Comments 6: The approach has some novelty in it, but it got lost in the long tedious tables of nearly same results of similar machine learning models working on the limited dataset that perform with nearly equal quality metrics.
Response 6: Thank you for your thoughtful feedback. We intended to provide a comprehensive evaluation in presenting results across multiple settings and demonstrate that our proposed framework consistently achieves competitive classification performance, even when using fewer features or smaller subsets of data. This suggests the potential for reducing clinical data collection and processing time, which is an important practical benefit in real-world settings.
To improve readability and avoid overwhelming the reader with repetitive details, we have moved several parts of the results section into the appendix (e.g. Tables A3-A4 on page 18 marked in red).
Round 2
Reviewer 2 Report
Comments and Suggestions for Authors
I would suggest simplifying the wording of the Introduction part a bit, it still has too many unnecessary superlative adjectives. The rest of the article has been significantly improved and can be accepted for publication after minor revision.
Author Response
We sincerely thank you for the time and effort spent reviewing our manuscript. In response to the comment, we have simplified the wording in the Introduction to reduce unnecessary superlative adjectives. Revisions are highlighted in blue with wavy underlines in the updated manuscript.